# Perception of COVID-19 Booster Dose Vaccine among Healthcare Workers in India and Saudi Arabia

**DOI:** 10.3390/ijerph19158942

**Published:** 2022-07-22

**Authors:** Sajith Vellappally, Sachin Naik, Omar Alsadon, Abdulaziz Abdullah Al-Kheraif, Haya Alayadi, Areej Jaber Alsiwat, Aswini Kumar, Mohamed Hashem, Nibu Varghese, Nebu George Thomas, Sukumaran Anil

**Affiliations:** 1Dental Health Department, College of Applied Medical Sciences, King Saud University, Riyadh 11433, Saudi Arabia; oalsadon@ksu.edu.sa (O.A.); halayadi@ksu.edu.sa (H.A.); aalsiwat@ksu.edu.sa (A.J.A.); mihashem@ksu.edu.sa (M.H.); 2Dental Biomaterials Research Chair, Dental Health Department, College of Applied Medical Sciences, King Saud University, Riyadh 11433, Saudi Arabia; aalkhuraif@ksu.edu.sa; 3Department of Prosthodontics, Amritha School of Dentistry, Vishwavidyapeetham, Cochin 682041, KL, India; aswinikumark@aims.amrita.edu; 4MACFAST, Tiruvalla 689101, KL, India; researchdirector@macfast.org; 5Department of Periodontology, Pushpagiri College of Dental Sciences, Thiruvalla 689107, KL, India; nebugeorgethomas@pushpagiri.in; 6Department of Dentistry, Oral Health Institute, Hamad Medical Cooperation, Doha 3050, Qatar; asukumaran1@hamad.qa; 7College of Dental Medicine, Qatar University, Doha 2713, Qatar

**Keywords:** COVID-19, COVID-19 vaccines, health personnel, awareness, vaccination hesitancy, India, Saudi Arabia

## Abstract

Background: COVID-19 vaccines were made available to the public by the end of 2020. However, little is known about COVID-19 booster dose (CBD) vaccine perception among healthcare workers (HCW) worldwide. The present study aims to assess the perception of CBD vaccines among healthcare workers in India and Saudi Arabia (SA). Methods: A cross-sectional study was conducted among HCWs in two countries, India and SA. Data were gathered through the use of a self-administered questionnaire. A convenience sampling technique was utilized to collect the data. Results: A total of 833 HCW responses were collected from the two countries, with 530 participants from India and 303 participants from SA responding to the questionnaire. Among them, 16% from India and 33% from SA were unwilling to take a CBD (*p* < 0.005). The primary reasons for not being willing were concerns about whether the vaccine would be effective (32%) and concerns about probable long-term side effects (31%). Concerns about not knowing enough about the vaccination (30%) and the possibility of long-term side effects (28%) were the primary concerns in SA. Regression analysis showed that males, urban residents, and post-graduates were more willing to take the CBD. Conclusion: There is a good perception of CBD and some hesitancy in receiving the booster dose among HCWs in both countries. The introduction of personalized education, risk communication, and deliberate policy could help to reduce the number of people who are unwilling to take a booster shot.

## 1. Introduction

The COVID-19 disease caused by the SARS-CoV-2 virus is a respiratory disease with symptoms ranging from asymptomatic to mild or severe complications, such as respiratory distress, pneumonia, and death [1]. Non-pharmaceutical interventions such as facemask wearing, quarantine, and social distancing have shown some effectiveness in containing the spread of the disease [2]. However, a safe vaccination program with broad clinical benefits is considered a suitable long-term solution when implemented globally. The vaccine must substantially reduce morbidity and mortality, which is beneficial for both healthcare workers and the public [3]. To achieve herd immunity and restrain the spread of the disease, extensive vaccination is one of the prerequisites [4]. Multiple COVID-19 vaccines have been developed, with the vaccination process accelerated in a few countries; however, some people are still uncertain regarding the efficacy, dosage, and safety of the vaccine [5]. Furthermore, post-vaccination adverse events have been reported, ranging from mild complications to death and inactivated vaccines, and the number of new variants identified worldwide has been increasing [6,7]. Acceptance and demand for the vaccination process are multi-factorial, attributed to misinformation, and vary across time and place, perception of the risks and disease, culture, religion, and other reasons [8]. One study in the United Arab Emirates has shown that too many injections were also a key factor for vaccine hesitancy [9].

The data that are currently available have demonstrated that the vaccines that are currently available are effective against the COVID-19 disease, although for a limited period [10]. Studies have shown that the effect of protection from vaccines against symptomatic disease gradually tapers over a period, thus leading to the demand for a COVID-19 booster dose (CBD) [11], with a substantial increase in protection against symptomatic COVID-19 disease following a booster dose [12]. The disease severity is reduced, ranging from 92–97% [13], and there is a reduction in testing positive for SARS-CoV-2 among the vaccinated [14]. The ideal time to accomplish the best assurance against SARS-CoV-2-related results after the booster dose is still unknown; however, it has been found that the effectiveness appears seven days after the booster dose, as attributed to high antibody levels. A booster dose has been shown to reduce the infection rate in recipients by over a factor of ten, meaning that the susceptibility to infection would decrease approximately by 5%, compared to unvaccinated individuals [15]. An increase in antibodies has been observed after 3–5 days following the administration of a booster dose, unlike the influenza virus vaccine, where it was seen immediately (i.e., in the following two days) [16]. Post-exposure to the coronavirus following two doses has also been shown to prevent infections, termed a post-exposure effect [17].

Healthcare workers have higher infection rates than other professions [18]. The Centers for Disease Control and Prevention has reported that more than 1600 US healthcare workers have died from COVID-19 [19]. Vaccine mandates are expected to prevent infections, serious illness, and deaths in healthcare workers, regardless of where they are exposed. Healthcare professionals play a crucial part in the effectiveness of immunization campaigns. According to studies, their knowledge and attitude regarding vaccines are essential to the success of immunization campaigns. It has been demonstrated that their vaccine knowledge and attitudes determine their vaccine uptake intentions and recommendations to larger groups [20,21,22].

Vaccination or immunization programs are beneficial only when there is a higher rate of perseverance and acceptance by the target population. Evidence has suggested that a substantial proportion of fully vaccinated members of the general public are hesitant to get a COVID-19 vaccination booster dose [23,24] HCWs, being a defined high-risk category for COVID-19 infection, were prioritized in all nations that rolled out the COVID-19 booster dosage vaccination. This study describes the acceptance of a booster dose among healthcare workers, the most vulnerable population. Furthermore, we aim to provide relevant information to the public health authorities, in order to prevent the disease from spreading further. Questionnaires are helpful instruments for determining the level of comprehension and motivation regarding vaccination. This study was conducted to determine how people in India and Saudi Arabia feel about COVID-19 booster doses. These two countries are very different, regarding their culture and economy, so their perspectives on the COVID-19 booster dose may differ.

## 2. Materials and Methods

### 2.1. Study Design and Participants

A cross-sectional study was carried out in India and Saudi Arabia. Data were collected using a convenience sampling method between 31 January and 10 March 2022. The information was gathered from a sample of HCWs in both counties. The inclusion criteria were HCWs (defined by the WHO International categorization of health workers) [25] and individuals who expressed an interest in participating. Subjects having a history of mental illness were excluded from participating in the study. Ethical approval was granted by College of Applied Medical Sciences, King Saud University. We received responses from 833 participants from both nations.

### 2.2. Study Procedure

The data were gathered through a self-administered questionnaire distributed through internet messaging services, such as WhatsApp and email (see Appendix A). Participants were requested to provide informed consent before being included in the study.

### 2.3. Measures

The questionnaire was developed based on previous studies [26,27] and frequently asked questions (FAQ) by the World Health Organization (WHO) [28], Centers for Disease Control and Prevention (CDC) website [29]. After administering the questionnaire to a panel of ten dental professors, administrators, and experts to determine its validity, necessary modifications were made. After a validity check, a kappa value of 0.70 was determined.

The questionnaire was divided into two sections: (1) Demographic information and their willingness to get a CBD vaccination; and (2) information on factors influencing their perception of taking a CBD vaccination. We conducted a pilot study in a small group of participants (*n* = 50), who were asked to contribute their ideas on simplifying and shortening the questionnaire. We recruited participants from a variety of socio-economic backgrounds for the pilot study. The comments made by participants were considered and integrated into the survey. After a thorough discussion, the authors accepted the questionnaire (20 items) and circulated it to conduct the study. It took each participant around 2–3 min to complete the questionnaire.

### 2.4. Socio-Demographic and Past Vaccination History Information

Personal information, including residence, age, gender, and education, was obtained. Participants were questioned about their immunization history during their lifetime, including two doses of the COVID-19 vaccine and the type of company from whom they had taken the vaccine.

### 2.5. COVID-19 Booster Vaccine Perception

After their willingness to take the CBD, participants were questioned about their perception regarding the CBD. These questions were graded on a two-point scale, with the answers being “Yes” and “No.”

### 2.6. Statistical Analysis

Descriptive statistics were used to characterize the sociodemographic information of the people from India and SA. The chi-square test was used to evaluate CBD perceptions of the participants. The binary logistic regression results were compared between India and SA, in order to determine how the socio-demographic components were perceived. In this study, the adjusted odds ratio was calculated using a confidence interval of 95%. The statistical software IBM-SPSS version 25 (BM Corporation, Armonk, NY, USA) was used for the analysis.

## 3. Results

### 3.1. Participants (Demographics)

The demographic information of participants is depicted in Table 1. It can be seen that 74% of Indian participants were between the ages of 21 and 30, compared to 71% of Saudi Arabian participants. India had 37% of its participants living in rural areas, whereas SA had 47%.

### 3.2. Willingness to Take CBD

In India, 84% of individuals were willing to receive the vaccination, while only 64% in Saudi Arabia were (*p* < 0.05; Figure 1). This revealed that most participants in both countries were willing to receive the CBD. In India, 97% of the HCWs had received two doses of the COVID 19 vaccination, compared to 79% in Saudi Arabia.

### 3.3. Reasons for Not Willing to Take CBD

A total of 16% of respondents in India and 33% in Saudi Arabia were unwilling to take the CBD. In India, apprehensions regarding the vaccine’s efficacy (32%) and possible long-term adverse effects were the leading impediments to vaccination (31%). Concerns about not understanding enough about the immunization (30%) and the likelihood of long-term adverse effects (28%) were the most prevalent reasons in SA (Figure 2).

### 3.4. Perception about CBD

The perceptions of CBD among the participants are presented in Table 2. COVID-19 vaccination was considered unsafe by 17% of individuals from India and 32% of participants from SA, while 14% of participants from India and 44% in SA believe that the CBD has adverse side effects. A total of 420 respondents from India and 191 from SA stated that they would encourage their family, friends, and relatives to take the CBD vaccination. Participants from India (63%) and SA (60%) believed that the CBD can help to slow the spread of COVID-19, while 58% of respondents from both India and SA believed that pharmaceutical companies have developed safe and effective COVID-19 vaccines. Then, 28% of those surveyed in India and 56% in SA believed they were given the COVID-19 dosage because it was mandatory. Regarding the mixing and matching of booster doses, 84% of Indians and 51% of Saudis did not think that it is safe and effective.

Table 3 shows the results of a binary logistic regression analysis considering willingness to take the CBD in either a “Yes” or “No” answer. Statistical significance was found for India, with X^2^(115) = 230 (*p* < 0.005), and SA, with X^2^(72) = 144 (*p* < 0.005), in the logistic regression model. In India, the model explained 60% (Nagelkerke R2) of the variance in willingness to receive the CBD and correctly classified 91.1%. In SA, the model explained 52% (Nagelkerke R2) of the variance and correctly classified 82.5%.

In addition, Table 3 details the characteristics that influenced perceptions of participants regarding the COVID-19 booster dose. Age had an odds ratio (OR) of 1.10 in India and 1.08 in SA, indicating that older participants were more likely to accept the CBD. Males were more likely to receive the vaccination (OR 1.21 in India, 1.47 in KSA), and those who resided in urban areas (OR 0.65 in India, 1.38 in SA) were also more willing to do so. In response to the question regarding the detrimental effects of CBD, the OR for respondents of both countries was more than 1. For the question of whether mixing and matching booster doses is safe and effective, the OR was 1.16 for India and 1.59 for Saudi Arabia. Regarding the question of whether a booster vaccine prevents the spread of COVID-19, India and Saudi Arabia had ORs of 0.82 and 0.86, respectively.

## 4. Discussion

The perceptions and attitudes of HCWs concerning CBD vaccination significantly impact the public’s attitude towards it. There are 5.76 million HCWs in India and 1.12 million HCWs in SA [30,31,32]. The heterogeneity of participation from both genders, age groups, and various healthcare categories, as well as the involvement of HCWs dealing with COVID-19 patients, makes the present study interesting. This study is intended to guide both country health authorities and public health professionals regarding the COVID-19 vaccination program. According to a review, the proportion of HCWs who were hesitant to take the COVID-19 vaccination ranged from 4.3% to 72% [33]. We observed that 84% of Indian and 67% of Saudi Arabian healthcare workers were willing to receive the booster vaccination. A cross-national study regarding vaccination hesitancy and preferences between the United States (U.S.) and China has revealed that both nations had high acceptability of vaccination, despite the vaccination policies [4]. Only 23% of healthcare workers were willing to receive vaccination in a Taiwanese study during the COVID-19 pandemic [34]. In a similar survey carried out in Egypt, 51% were undecided regarding whether to be vaccinated [35]. A survey was done to determine the willingness of HCWs in SA to get the COVID-19 vaccination. The survey included HCWs from all administrative regions of the country. Of the 1124 HCWs that participated in the survey, 674 responded and 35.1% expressed a reluctance to receive the vaccination [36].

Even though the two considered countries—India and SA—have a high vaccine acceptability level, significant differences were observed. This may be due to the differences in the COVID-19 prevalence, the availability of COVID-19 vaccines, and vaccine policies [37]. A study of healthcare workers in two Indian hospitals assessed attitudes and willingness to accept the COVID-19 vaccination. Among the 520 participants who responded to the survey, 63% indicated they were willing to be vaccinated, while 46% of dental and 49% of medical HCWs exhibited vaccine reluctance [38]. A similar study was conducted on HCWs working in a tertiary care center in India; 54% of those surveyed were unaware of the determinants of the COVID vaccine, 20% exhibited reluctance, and 18% indicated strong opposition to the vaccine [39]. The present survey revealed that, in India, apprehension that the vaccination would not be successful was the top reason for the reluctance to receive it (32%), whereas Saudi Arabian HCWs stated a lack of sufficient knowledge regarding the vaccine (30%). According to the CDC, responses following a booster injection have been comparable to those reported after administering two or more vaccination doses. The majority of adverse effects were mild to moderate in intensity. Although serious side effects are uncommon, they can occur [40].

Our survey also found that 17% of Indian and 32% of Saudi Arabian participants felt that COVID-19 vaccinations are unsafe. In a survey among 1068 medical students from 22 states in India to determine the reasons for vaccine reluctance, 11% of participants showed vaccine hesitation due to safety concerns. Concerns about the safety and efficacy of vaccinations and a lack of confidence in government organizations added to this reluctance [41]. The availability of different vaccine brands in the two countries and the manufacturer claims regarding vaccine safety might have affected vaccine acceptability. In our survey, 14% of individuals in India and 44% in Saudi Arabia believed that the COVID-19 vaccine causes adverse side effects. As more information about COVID-19 is being disseminated via social media and news networks, it is crucial that it is technically authentic and scientifically validated, such that neither the recipients nor the general public are influenced. Several researchers have investigated the reasons for vaccine hesitancy in various demographics, particularly among healthcare workers, for the influenza vaccine. The reasons for vaccine reluctance were: (1) Inadequate awareness campaigns; (2) modified perceived risks; (3) inadequate education on the efficiency of the influenza vaccine and its potential adverse reactions, (4) lack of access to vaccination centers; and (5) socio-demographic factors [42].

In the current survey, 79% of Indian and 73% of Saudi Arabian participants advised their family and friends to take a CBD. In a study conducted in the Czech Republic, 83% of participants indicated that the safety of their family was their primary reason for accepting a CBD, followed by self-protection (82.7%) [43]. Many governments and organizations have implemented mandatory vaccination for people working in specific fields, such as healthcare, transport, education, and retail, in order to minimize the spread of COVID-19. In our study, most participants responded that they had received the CBD as it was mandatory. Tajikistan was the first country to mandate that all people older than 18 receive the COVID-19 vaccine. Saudi Arabia has mandated that all Hajj pilgrims and those seeking to attend private or government facilities be fully vaccinated [44].

Although most Indian HCWs (84%) stated that mixing the booster dose is ineffective in preventing disease, 49% of Saudi HCWs believed it to be effective, even when brands were mixed. A similar study was undertaken in the KSA, in order to determine HCW perceptions regarding the novel Delta variant and their readiness to receive a third vaccine dose, where 64% of respondents believed that mix-matching would be successful. The subjects reported considerable reluctance and a moderate willingness to receive a vaccine booster dose [45]. Most vaccination skeptics argue that vaccine manufacturers have not created effective vaccines. mRNA-based vaccines, attenuated live vaccines, viral vectors, and other types of vaccine are now being developed by pharmaceutical companies worldwide [46].

In our study, logistic regression showed that participants who were willing to receive a CBD believed that it has adverse reactions, that mix-matching the booster dose is safe and effective, and that the booster vaccine can reduce the spread of COVID-19. In Singapore, 98.9% of HCWs were completely immunized, and 73.8% of those eligible had taken a booster dose, which helped to decrease community spread and keep the mortality rate low [47].

The present study is limited by the fact that it is cross-sectional, which means that we cannot provide information about changes in CBD perception over time. Another factor is that we did not classify the healthcare workers based on their category. As the research was conducted in two countries with different cultures and economies, its conclusions may not be applied universally.

## 5. Conclusions

This survey provides valuable information regarding COVID-19 vaccine booster hesitancy and the potential variables influencing it. The relatively high booster vaccine acceptability among HCWs could result from earlier COVID-19 vaccination experience, regarding safety, and the high risk of contracting COVID-19 from healthcare facilities. Concerns about vaccine safety, vaccine efficacy, and lack of trust were possible underlying causes of vaccine hesitancy. HCWs are positively influenced by close friends and co-workers who value COVID-19 vaccination, which may encourage the development of cross-departmental interactions to increase vaccination rates. The presented observations and conclusions may serve as tools for building future policies and public health actions designed to increase the COVID-19 vaccination rate.

## Figures and Tables

**Figure 1 ijerph-19-08942-f001:**
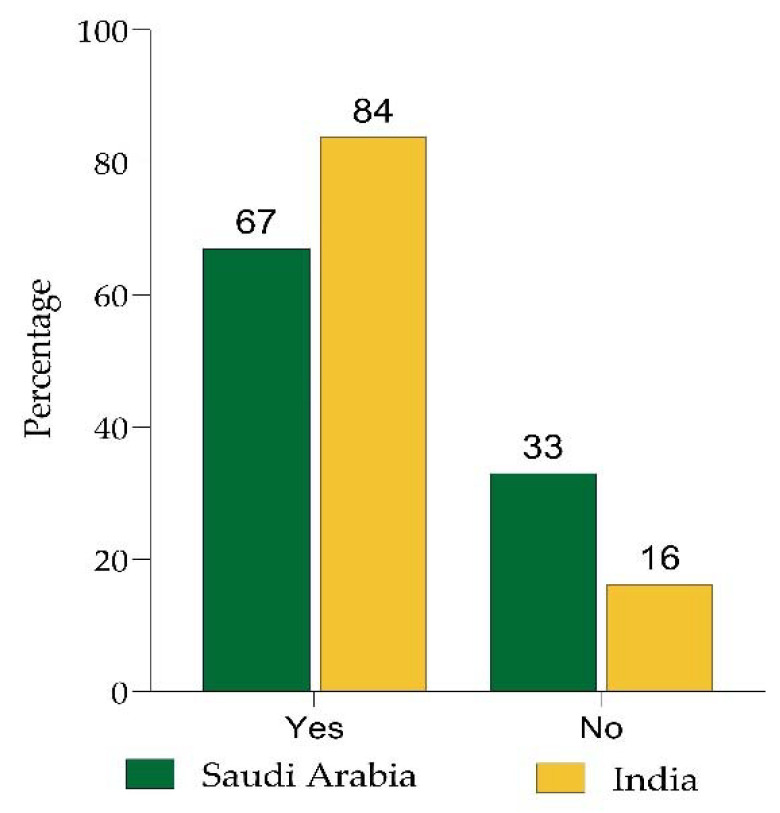
HCWs willing to take the COVID-19 booster vaccine without any hesitation.

**Figure 2 ijerph-19-08942-f002:**
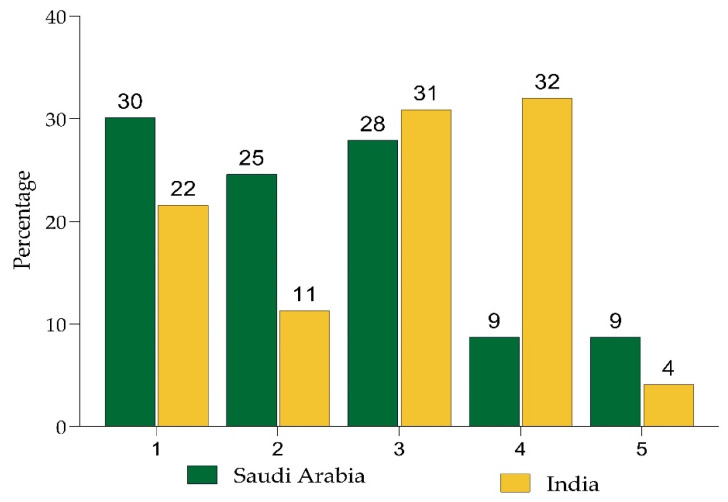
Reasons for not being willing to take a COVID-19 booster dose vaccine. 1. I am concerned as I do not know enough about the vaccine; 2. I am concerned about the short-term side effects (e.g., fever etc.); 3. I am concerned about possible long-term side effects; 4. I am concerned because I do not think the vaccine will be effective; 5. I am against vaccines in general.

**Table 1 ijerph-19-08942-t001:** Demographic information of participants, as a major source of information and acceptance.

Demographic Items	India		Saudi Arabia		*p*-Value
	Number (%)		Number (%)		
**Sample size, *n***	530 (64)		303 (36)		
Gender					
Male	154 (29)		219 (72)		<0.001 *
Female	376 (71)		84 (28)		
**Age interval (in years)**					
21–30	392 (74)		214 (71)		0.333
31–40	66 (12)		46 (15)		
>40	72 (14)		43 (14)		
**Highest educational level**					
Undergraduate	413 (78)		137 (45)		<0.001 *
Post-graduate	100 (19)		123 (41)		
PhD	17 (3)		43 (14)		
**Residence**					
Rural	197 (37)		143 (47)		<0.001 *
Urban	333 (63)		160 (53)		
Have you received all the necessary vaccines in your lifetime?	Yes	No	Yes	No	0.794
	410 (77%)	120 (23%)	232 (77%)	71 (23%)	
Do you know about the COVID-19 vaccine?	521 (98)	9 (2)	256(84)	47(16)	<0.001 *
Have you received two doses of COVID-19 vaccines?	514 (97)	16 (3)	241 (79)	62 (21)	<0.001 *
Are you willing to take the COVID-19 booster vaccine or take it without any hesitation?	444 (84)	86 (16)	203 (67)	100 (33)	<0.001 *

* Significant.

**Table 2 ijerph-19-08942-t002:** Perception about COVID 19 booster dose among healthcare workers.

Questions	India		Saudi Arabia		*p* Value
	Number (%)		Number (%)		
	Yes	No	Yes	No	
Do you believe that the COVID-19 vaccine is safe?	442 (83)	88 (17)	206 (68%)	97 (32%)	<0.001 *
Do you think that COVID-19 booster vaccination has adverse reactions?	76 (14)	454 (86)	135 (44)	168 (56)	<0.001 *
Do you encourage your family/friends/relatives to get the booster COVID-19 vaccine?	420 (79)	110 (201)	191 (73)	112 (27)	<0.001 *
Do you believe the COVID-19 booster vaccine can reduce the spread of COVID-19?	335 (63)	195 (367)	183 (60)	120 (40)	0.421
Do you believe the COVID-19 booster vaccine can reduce the complications associated with COVID-19?	387 (73)	143 (27)	207 (68)	96 (32)	0.149
Do you think that if everyone in society maintains the preventive measures, the COVID-19 pandemic can be eradicated without vaccination?	200 (38)	330 (62)	147 (48)	156 (52)	0.002*
Do you think Pharmaceutical companies have developed safe and effective COVID-19 vaccines?	307 (58)	223 (42)	175 (58)	128 (42)	0.926
Have you received COVID- 19 booster dose because it is mandatory?	150 (28)	380 (72)	169 (56)	134 (44)	<0.001 *
Do you think Mix-Matching the booster dose is safe and effective?	87 (16)	443 (84)	148 (49)	155 (51)	<0.001 *
Do you believe that only high-risk individuals such as health care workers and elderly persons with other diseases only need a booster dose?	94 (18)	436 (82)	146 (48)	157 (52)	<0.001 *

* *p*-value < 0.05.

**Table 3 ijerph-19-08942-t003:** Binomial logistic regression analysis exploring factors associated with willingness to take the COVID-19 booster vaccine in India and Saudi Arabia.

	India				Saudi Arabia			
			95% Confidence Interval for OR				95% Confidence Interval for OR	
	Sig.	OR	Lower	Upper	Sig.	OR	Lower	Upper
Age	0.003	1.10	0.61	1.99	0.749	1.08	0.66	1.77
Sex (Male)	0.001	1.21	0.54	2.72	0.338	1.47	0.67	3.21
Education	0.147	0.51	0.20	1.27	0.481	1.19	0.73	1.95
Residence (Urban)	0.001	0.65	1.31	1.37	0.363	1.38	0.69	2.82
Do you think that COVID-19 booster vaccination has adverse reactions? (Yes)	0.285	1.59	0.68	3.73	0.604	0.83	0.42	1.66
Do you encourage your family/friends/relatives to get the booster COVID-19 vaccine? (Yes)	0	0.03	0.01	0.06	0	0.13	0.06	0.28
Do you believe that the COVID-19 booster vaccine can reduce the spread of COVID-19? (Yes)	0.653	0.82	0.35	1.93	0.687	0.86	0.41	1.80
Do you believe the COVID-19 booster vaccine can reduce the complications associated with COVID-19? (Yes)	0.244	0.61	0.26	1.41	0	0.21	0.10	0.45
Do you think that if everyone in society maintains the preventive measures, the COVID-19 pandemic can be eradicated without vaccination? (Yes)	0.394	0.73	0.35	1.51	0.623	1.19	0.59	2.41
Do you think Pharmaceutical companies have developed safe and effective COVID-19 vaccines? (Yes)	0.073	0.50	0.24	1.07	0.009	0.38	0.18	0.79
Have you received COVID-19 booster dose because it is mandatory? (Yes)	0.013	0.32	0.13	0.79	0.164	0.60	0.29	1.24
Do you think Mix-Matching the booster dose is safe and effective? (Yes)	0.001	1.16	0.35	3.81	0.19	1.59	0.27	1.29
Do you believe that only high-risk individuals such as health care workers and elderly persons with other diseases only need a booster dose? (Yes)	0.055	2.28	0.98	5.27	0.147	1.77	0.82	3.82
Constant	0	57.61			0	11.51		

OR, odds ratio; df, degrees of freedom; Sig, significance.

## Data Availability

The relevant data are contained within the article.

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
