# Peer review of "Perception of COVID-19 Booster Dose Vaccine among Healthcare Workers in India and Saudi Arabia"

_ijerph, 2022, doi:10.3390/ijerph19158942_

Round 1

Reviewer 1 Report

Dear editor,

Thank you for the invitation to review this manuscript. Attached below are my comments for the authors' consideration. 

Keywords

- Suggest to use MESH terms

Introduction

- The authors need to explain the need on why HCWs were the focus of this study as there were limited description in the introduction

-> Lesser focus should be made with regards to the efficacy of booster doses as this is not the focus of this manuscript. 

- The introduction will also benefit from a brief introduction about general levels of COVID-19 vaccine hesitancy and factors affecting vaccine hesitancy. 

-> Citations for consideration: https://pubmed.ncbi.nlm.nih.gov/344520; https://pubmed.ncbi.nlm.nih.gov/25896383/#:~:text=Abstract,across%20time%2C%20place%20and%20vaccines.

Methods

- A copy of the questionnaire should be attached as a supplementary material.

- Some of the questions appear oddly phrased 

-> "Do you know about the COVID-19 vaccine?"

--> It would have been better phrased as "Are you aware about COVID-19 vaccine" 

-> "Do you think that COVID-19 booster vaccination has adverse reactions?"

--> this appears quite similar to the question if COVID-19 vaccine is afe

- Were both the internal and external validity of the questionnaire evaluated? 

-> If so, how was it assessed. 

- Another potential issue with the questionaire was that only binary options were made available. 

Results

-> The occupation of the study population should be reflected in Table 1 as it would give a better representation of the HCW population

-> Table 3 - no need to report beta, SE, DF if the odds ratio is reported.

- Line 169 to 189 will beneft from tidying up. 

-> There is no need to explain how to interpret an odds ratio.

-> Readiness to receive booster and vaccine needs to be standardise as this study is specifically evaluating readiness to receive booster vaccine.

- For figure 2, are the participants allowed multiple answers? 

-> The appropriate denominator should be reported if so. Otherwise using a bar chart may be more appropriate for the presentation of results. 

Discussion

- The authors should consider discussing their results with regards to other Asian countries in the region such as Singapore and Taiwan

-> https://pubmed.ncbi.nlm.nih.gov/35455286/

-> https://www.ncbi.nlm.nih.gov/pmc/articles/PMC8000386/

- There is a need to highlight new/unique findings from this study as quite extensive work has been performed in this aspect related to vaccine hesitancy in HCWs globally.

Conclusion

- Do the authors have any cut-offs for defining good perception for COVID-19 booster as a significant proportion of HCWs in Saudia arabia were vaccine hesitant.

- There are 2 points in the conclusion which did not reflect what the study results showed.

-> Vaccine hesitancy amongst HCW has decreased from booster to first and second dose. Personalized education, risk communication, and deliberate policy could all help reduce the number of people who miss their booster shots

-> The above 2 points were not evaluated in this study and it would be inaccurate to conclude the above.

Language errors

1. Multiple grammatical errors exist within the manuscript

- examples of sentence structures that require improvement include

-> "Present data have shown high levels of a short period of protection by the currently 55

available COVID-19 vaccines against the disease"  

-> it demands making the HCW perceptions of COVID-19, recognizing the importance of the CBD

2. There are also use of odd phrasing e.g. massive vaccination 

-> Do the authors mean vaccination of masses? or mass vaccination?

Author Response

Response to Reviewers Suggestions/Comments

Manuscript ID: ijerph-1784105

Title: Perception of COVID-19 booster dose vaccine among healthcare workers in India and Saudi Arabia

Authors: Sajith Vellappally *, Sachin Naik *, Omar Alsadon, Abdulaziz Abdullah Al-Kheraif, Haya Alayadi, Areej Jaber Alsiwat, Aswini Kumar, Mohamed Hashem, Nibu Varghese, Nebu George Thomas, Sukumaran Anil

REVIEW REPORT FORM

REVIEWER 1

Comments

Response

Keywords: Suggest using MESH terms

Keywords selected based on MeSH terms

COVID-19; COVID-19 Vaccines; Health Personnel; Awareness; Vaccination Hesitancy; India; Saudi Arabia

Introduction

The authors need to explain the need on why HCWs were the focus of this study as there were limited description in the introduction.

Health care workers have higher infection rates than other professions [1]. The Centers for Disease Control and Prevention reported that more than 1,600 US health care workers have died from COVID-19 [2]. Vaccine mandates will prevent infections, serious illness, and deaths in health care workers regardless of where they are exposed. Healthcare professionals play a crucial part in the effectiveness of immunization campaigns. According to studies, their knowledge and attitude regarding vaccines are essential to the success of immunization campaigns. It has been demonstrated that their vaccine knowledge and attitudes determine their vaccine uptake intentions and recommendations to larger groups [3,4].

Lesser focus should be made with regards to the efficacy of booster doses as this is not the focus of this manuscript. 

We have made changes throughout the manuscript taking into consideration this point

The introduction will also benefit from a brief introduction about general levels of COVID-19 vaccine hesitancy and factors affecting vaccine hesitancy. 

Vaccine hesitancy is the delay in vaccine acceptance or refusal despite the availability of immunization services. In addition, it is not an exclusive phenomenon, meaning that some individuals will refuse a Covid-19 vaccine if offered, while others are dubious of their vaccination intentions [5]. Vaccine reluctance is a global concern and one of the most important causes of under-vaccination. According to the World Health Organization, vaccine hesitancy is one of the top ten global health threats and a significant barrier to the effectiveness of immunization programs [6]. Interestingly, negative information about the vaccination disseminated on social media also may have contributed to vaccine reluctance [7].

Citations for consideration: https://pubmed.ncbi.nlm.nih.gov/344520;

Verger P, Fressard L, Collange F, et al. Vaccine Hesitancy Among General Practitioners and Its Determinants During Controversies: A National Cross-sectional Survey in France. EBioMedicine. 2015;2(8):891-897.

METHODS

A copy of the questionnaire should be attached as supplementary material.

A copy of the questionnaire is attached as Annex 1

Some of the questions appear oddly phrased 

·     "Do you know about the COVID-19 vaccine? "It would have been better phrased as "Are you aware about COVID-19 vaccine" 

·     "Do you think that COVID-19 booster vaccination has adverse reactions?" this appears quite similar to the question if COVID-19 vaccine is safe

We agree with these comments. We thought of taking the main point from a negative and positive version; hence we use these two questions.

Were both the internal and external validity of the questionnaire evaluated? 

·     If so, how was it assessed? 

Several similar studies tested all the questions used in this survey. The external validity we rechecked by conducting a pilot study.

“The questionnaire was developed based on previous studies [8,9] and frequently asked questions (FAQ) by the World Health Organization (WHO)[10], Centers for Disease Control and Prevention (CDC) website [11]. After administering the questionnaire to a panel of ten dental professors, administrators, and experts to determine its validity, the necessary modifications were made. After a check for validity, the kappa value of 0.70 was determined.”

The questionnaire was divided into two sections: (1) demographic information and their willingness to get a CBD vaccination. (2) Contained information on factors influencing people's perceptions of taking a CBD vaccination. We conducted a pilot study with a small group of participants (n = 50), who were asked to contribute their ideas on simplifying and shortening the questionnaire. We recruited participants from a variety of socioeconomic backgrounds for the pilot research. The comments made by participants were considered and integrated into the survey. After a thorough discussion, the authors accepted the questionnaire (20 items) and circulated it to conduct the study. It took each participant around 2- 3-minutes to complete the task.

Another potential issue with the questionnaire was that only binary options were available. 

The main objective of the present survey is to identify the vaccine hesitancy among health care workers in India and Saudi Arabia and to find any significant differences in the perception of vaccine booster doses in these two counties. We use a binary questionnaire to simplify the process.

A comparative assessment of the two types of questiore survey revealed that "ordinally scaled questionnaires do not differ sufficiently in terms of profile interpretation to justify the use of such scales in preference to binary scales, which prove to be perceived as more difficult by respondents, and objectively take more time to complete." [12]

RESULTS

The occupation of the study population should be reflected in Table 1 as it would give a better representation of the HCW population.

Unfortunately, we did not categorize the health care workers based on their classification or nature of work. It is pooled data from all HCW categories based on the World Health Organization. As suggested, it would have given more insights.

Table 3 - no need to report beta, SE, or DF if the odds ratio is reported. There is no need to explain how to interpret an odds ratio. Line 169 to 189 will benefit from tidying up. 

beta, SE, DF removed from the table

Readiness to receive booster and vaccine needs to be standardize as this study is specifically evaluating readiness to receive booster vaccine. For figure 2, are the participants allowed multiple answers? 

No. Only one answer from the five variables

The appropriate denominator should be reported if so. Otherwise using a bar chart may be more appropriate for the presentation of results. 

Changed to a bar chart with values

Discussion

The authors should consider discussing their results with regards to other Asian countries in the region such as Singapore and Taiwan-> https://pubmed.ncbi.nlm.nih.gov/35455286/

There is a need to highlight new/unique findings from this study as quite extensive work has been performed in this aspect related to vaccine hesitancy in HCWs globally.

The discussion section was rewritten, considering the major focus areas. The surveys conducted in Asia were also included and compared [13].

Kukreti, S.; Lu, M.Y.; Lin, Y.H.; Strong, C.; Lin, C.Y.; Ko, NY; Chen, P.L.; Ko, WC Willingness of taiwan's healthcare workers and outpatients to vaccinate against covid-19 during a period without community outbreaks. Vaccines (Basel) 2021, 9.

Koh, S.W.C.; Tan, H.M.; Lee, W.H.; Mathews, J.; Young, D. Covid-19 vaccine booster hesitancy among healthcare workers: A retrospective observational study in singapore. Vaccines (Basel) 2022, 10.

Conclusion

Do the authors have any cut-offs for defining good perception for COVID-19 booster as a significant proportion of HCWs in Saudi Arabia were vaccine hesitant.

We could not determine a cut-off point based on the systematic reviews and available literature [14,15]. However, we believe that more than 50 percent acceptance is a logical acceptance level.

There are 2 points in the conclusion which did not reflect what the study results showed.

Vaccine hesitancy amongst HCW has decreased from booster to first and second dose. Personalized education, risk communication, and deliberate policy could all help reduce the number of people who miss their booster shots

The conclusion section is rewritten based on the outcome of our survey. 

The above 2 points were not evaluated in this study and it would be inaccurate to conclude the above.

Language errors

Multiple grammatical errors exist within the manuscript

examples of sentence structures that require improvement include

"Present data have shown high levels of a short period of protection by the currently 55 available COVID-19 vaccines against the disease"  

it demands making the HCW perceptions of COVID-19, recognizing the importance of the CBD. There are also use of odd phrasing e.g. massive vaccination 

Do the authors mean vaccination of masses? or mass vaccination?

Language editing is done.

References

  1. Shah, A.S.V.; Wood, R.; Gribben, C.; Caldwell, D.; Bishop, J.; Weir, A.; Kennedy, S.; Reid, M.; Smith-Palmer, A.; Goldberg, D., et al. Risk of hospital admission with coronavirus disease 2019 in healthcare workers and their households: Nationwide linkage cohort study. BMJ 2020, 371, m3582.
  2. Bandyopadhyay, S.; Baticulon, R.E.; Kadhum, M.; Alser, M.; Ojuka, D.K.; Badereddin, Y.; Kamath, A.; Parepalli, S.A.; Brown, G.; Iharchane, S., et al. Infection and mortality of healthcare workers worldwide from covid-19: A systematic review. BMJ Glob Health 2020, 5.
  3. Wagner, A.L.; Masters, N.B.; Domek, G.J.; Mathew, J.L.; Sun, X.; Asturias, E.J.; Ren, J.; Huang, Z.; Contreras-Roldan, I.L.; Gebremeskel, B., et al. Comparisons of vaccine hesitancy across five low- and middle-income countries. Vaccines (Basel) 2019, 7.
  4. Verger, P.; Fressard, L.; Collange, F.; Gautier, A.; Jestin, C.; Launay, O.; Raude, J.; Pulcini, C.; Peretti-Watel, P. Vaccine hesitancy among general practitioners and its determinants during controversies: A national cross-sectional survey in france. EBioMedicine 2015, 2, 891-897.
  5. MacDonald, N.E. Vaccine hesitancy: Definition, scope and determinants. Vaccine 2015, 33, 4161-4164.
  6. WHO. Who ten threats to global health in 2019. https://www.who.int/news-room/spotlight/ten-threats-to-global-health-in-2019 (July 2022),
  7. Dhaliwal, D.; Mannion, C. Antivaccine messages on facebook: Preliminary audit. JMIR Public Health Surveill 2020, 6, e18878.
  8. Hawlader, M.D.H.; Rahman, M.L.; Nazir, A.; Ara, T.; Haque, M.M.A.; Saha, S.; Barsha, S.Y.; Hossian, M.; Matin, K.F.; Siddiquea, S.R., et al. Covid-19 vaccine acceptance in south asia: A multi-country study. Int J Infect Dis 2022, 114, 1-10.
  9. Reiter, P.L.; Pennell, M.L.; Katz, M.L. Acceptability of a covid-19 vaccine among adults in the united states: How many people would get vaccinated? Vaccine 2020, 38, 6500-6507.
  10. WHO. Coronavirus disease (covid-19): Vaccines. https://www.who.int/news-room/questions-and-answers/item/coronavirus-disease-(covid-19)-vaccines (13July 2022),
  11. CDC. Covid vaccines: Frequently asked questions. https://www.cdc.gov/coronavirus/2019-ncov/faq.html (13 July 2022),
  12. Dolnicar, S. In Simplifying three-way questionnaires - do the advantages of binary answer categories compensate for the loss of information?, 2003.
  13. Kukreti, S.; Lu, M.Y.; Lin, Y.H.; Strong, C.; Lin, C.Y.; Ko, N.Y.; Chen, P.L.; Ko, W.C. Willingness of taiwan's healthcare workers and outpatients to vaccinate against covid-19 during a period without community outbreaks. Vaccines (Basel) 2021, 9.
  14. Larson, H.J.; Clarke, R.M.; Jarrett, C.; Eckersberger, E.; Levine, Z.; Schulz, W.S.; Paterson, P. Measuring trust in vaccination: A systematic review. Hum Vaccin Immunother 2018, 14, 1599-1609.
  15. Schmid, P.; Rauber, D.; Betsch, C.; Lidolt, G.; Denker, M.L. Barriers of influenza vaccination intention and behavior - a systematic review of influenza vaccine hesitancy, 2005 - 2016. PLoS One 2017, 12, e0170550.

.

Reviewer 2 Report

The cross-sectional study conducted by Sajith Vellappally and colleagues demonstrated the perception of CBD vaccine among 530 healthcare workers in India and 303 healthcare workers in Saudi Arabia using self-administered questionnaire. The study mainly detected the hesitancy in receiving the booster dose among the healthcare workers. Inference of willingness or hesitancy in taking the booster/ third dose of vaccine may be significant for the other front liners to endorse the necessity of taking the booster in order to get rid of the pandemic. Although the work lacks the immunological basis, still it’s important for the mass public awareness. However, I would like to suggest the authors to edit the language of the manuscript by any professional or native English speaker. While reading the paper, I felt many lines to be repeating or in excess.

Author Response

REVIEWER 2

The cross-sectional study conducted by Sajith Vellappally and colleagues demonstrated the perception of CBD vaccine among 530 healthcare workers in India and 303 healthcare workers in Saudi Arabia using self-administered questionnaire. The study mainly detected the hesitancy in receiving the booster dose among the healthcare workers. Inference of willingness or hesitancy in taking the booster/ third dose of vaccine may be significant for the other front liners to endorse the necessity of taking the booster in order to get rid of the pandemic. Although the work lacks the immunological basis, still it's important for the mass public awareness. However, I would like to suggest the authors to edit the language of the manuscript by any professional or native English speaker. While reading the paper, I felt many lines to be repeating or in excess.

Language editing is done, and there were few repetitions. During the editing process, we removed it.

Reviewer 3 Report

Paper highlight interesting question about vaccine hesitancy in HCW.

I would only suggest to better elucidate difference with other papers already published (https://doi.org/10.1016/j.ajic.2021.10.004) 

Further improve conslusion section

Author Response

REVIEWER 3

Comments and Suggestions for Authors

Paper highlight interesting question about vaccine hesitancy in HCW.

I would only suggest to better elucidate difference with other papers already published https://doi.org/10.1016/j.ajic.2021.10.004

Further improve conclusion section

The conclusion section was modified.

Through the manuscript on vaccine hesitancy among HCW, we followed the key message from the manuscript. We focused more on the third booster dose vaccine in the present survey.

Toth-Manikowski, S.M.; Swirsky, E.S.; Gandhi, R.; Piscitello, G. Covid-19 vaccination hesitancy among health care workers, communication, and policy-making. American Journal of Infection Control 2022, 50, 20-25.

Round 2

Reviewer 1 Report

Would suggest for further proof reading to correct grammatical errors and restructure some of the sentences in the manuscript

Author Response

Response to Reviewers Suggestions/Comments

Manuscript ID: ijerph-1784105

Title: Perception of COVID-19 booster dose vaccine among healthcare workers in India and Saudi Arabia

Authors: Sajith Vellappally *, Sachin Naik *, Omar Alsadon, Abdulaziz Abdullah Al-Kheraif, Haya Alayadi, Areej Jaber Alsiwat, Aswini Kumar, Mohamed Hashem, Nibu Varghese, Nebu George Thomas, Sukumaran Anil

REVIEW REPORT FORM

REVIEWER 1

COMMENTS

Response

Keywords: Suggest using MESH terms

The Appendix 1 – Changed as Supplementary Table 1.

The emails are correct.

Would suggest for further proof reading to correct grammatical errors and restructure some of the sentences in the manuscript

Language edited by MDPI-Editing Services-Certificate attached.

REVIEWER 2

Authors have improved the language issue.

Language edited by MDPI-Editing Services-Certificate attached.

.

Reviewer 2 Report

Authors have improved the language issue.

Author Response

(The authors gave the same response as above.)
